# Spermidine Rescues Bioenergetic and Mitophagy Deficits Induced by Disease-Associated Tau Protein

**DOI:** 10.3390/ijms24065297

**Published:** 2023-03-10

**Authors:** Lauren H. Fairley, Imane Lejri, Amandine Grimm, Anne Eckert

**Affiliations:** 1Research Cluster Molecular and Cognitive Neuroscience, University of Basel, 4002 Basel, Switzerland; 2Neurobiology Lab for Brain Aging and Mental Health, Psychiatric University Clinics, 4002 Basel, Switzerland; 3Department of Biomedicine, University of Basel, 4055 Basel, Switzerland

**Keywords:** tau, mitochondria, spermidine, bioenergetics, autophagy, mitophagy

## Abstract

Abnormal tau build-up is a hallmark of Alzheimer’s disease (AD) and more than 20 other serious neurodegenerative diseases. Mitochondria are paramount organelles playing a predominant role in cellular bioenergetics, namely by providing the main source of cellular energy via adenosine triphosphate generation. Abnormal tau impairs almost every aspect of mitochondrial function, from mitochondrial respiration to mitophagy. The aim of our study was to investigate the effects of spermidine, a polyamine which exerts neuroprotective effects, on mitochondrial function in a cellular model of tauopathy. Recent evidence identified autophagy as the main mechanism of action of spermidine on life-span prolongation and neuroprotection, but the effects of spermidine on abnormal tau-induced mitochondrial dysfunction have not yet been investigated. We used SH-SY5Y cells stably expressing a mutant form of human tau protein (P301L tau mutation) or cells expressing the empty vector (control cells). We showed that spermidine improved mitochondrial respiration, mitochondrial membrane potential as well as adenosine triphosphate (ATP) production in both control and P301L tau-expressing cells. We also showed that spermidine decreased the level of free radicals, increased autophagy and restored P301L tau-induced impairments in mitophagy. Overall, our findings suggest that spermidine supplementation might represent an attractive therapeutic approach to prevent/counteract tau-related mitochondrial impairments.

## 1. Introduction

Tauopathies are a group of diseases including Alzheimer’s disease (AD), frontal temporal dementia with Parkinsonism linked to chromosome 17 (FTDP-17), progressive supranuclear palsy, Pick’s disease and corticobasal degeneration. A typical hallmark of these diseases is formation of neurofibrillary tangles (NFTs) located in characteristic regional patterns [1]. NFTs are composed of abnormally hyperphosphorylated tau, a microtubule-associated protein expressed in most neurons. Tau protein normally binds to microtubules in neuronal axons, stabilizing their assembly and function [2,3]. Microtubules are not only essential neuronal structural elements; they are also involved in the intracellular transport of lipids, proteins, nucleic acids, synaptic vesicles and cell organelles such as mitochondria [3]. Axonal transport is essential for proper neuronal function as it provides the synapses with mitochondria, which are needed to meet the high energy requirements of this cellular compartment. Consequently, hyperphosphorylated tau causes fewer mitochondria to be transported to the synapses, which leads to decreased energy supply and synaptic degeneration [4].

Mitochondria are paramount organelles in cells, playing a critical role in neuronal homeostasis and function. Besides adenosine triphosphate (ATP) generation through oxidative phosphorylation (OXPHOS), mitochondria are involved in many other cellular functions, including cell growth and differentiation, apoptosis signaling, the regulation of intracellular calcium homeostasis, regulation of reduction-oxidation homeostasis, synaptic plasticity and synthesis of neurotransmitters [5,6]. Mitochondria are also the main source of reactive oxygen species (ROS) in the cell. ROS are useful to control cellular functions such as proliferation and differentiation, but they can also be harmful to mitochondria and the cell due to oxidative stress [7,8].

We, and others, have already shown that abnormal tau protein impairs almost every aspect of mitochondrial function, from mitochondrial bioenergetics, ROS generation and ATP levels, to mitochondrial quality control, also called mitophagy (reviewed in [3]). Namely, using a cellular model (SH-SY5Y cells) stably expressing the P301L tau mutation compared with control (healthy) cells, we previously showed that P301L-expressing cells exhibit mitochondrial respiratory deficits [9]. Specifically, we observed diminished mitochondrial complex I activity associated with a decreased ATP level in tau cells. We also showed that P301L cells exhibit other mitochondrial impairments, namely decreased maximal respiration and spare respiratory capacity, lower mitochondrial membrane potential and a decreased ability to synthesize neuroactive steroids [10,11]. Recently, the group of Jürgen Götz also demonstrated that P301L tau impacts the mitophagy process in mammalian cell culture models and *C. elegans* [12]. In line with this, studies using therapeutic strategies aimed at improving mitochondrial function have reported protective effects in modulating ATP production, improving neuronal survival and attenuating brain atrophy and neuroinflammation in pre-clinical models of tauopathies [10,11,13,14,15].

Spermidine is a polyamine synthesized from putrescine, and it serves as a precursor for spermine [16]. Studies have reported cardioprotective, neuroprotective and lifespan-promoting effects of this polyamine [17]. Autophagy has been identified as the main mechanism of action of spermidine on life-span prolongation [18,19]. Besides its role in inducing autophagy, spermidine was found to suppress the overproduction of ROS and the level of necrotic cell death, and was shown to reduce the damage from oxidative stress in aging mice [18,19]. Additionally, studies showed that spermidine exerts beneficial effects in neural aging associated with changes in mitochondrial structure and function following stress [20].

Based on these findings, we hypothesized that spermidine may exert protective effects on mitochondrial function in the presence of abnormal tau protein, via improving mitochondrial bioenergetics and mitophagy. We first aimed to evaluate the effect of spermidine on neuronal bioenergetics in P301L tau-expressing cells compared with control cells by measuring the cell metabolic activity, ATP level, mitochondrial membrane potential, mitochondrial respiration and ROS levels after spermidine treatment. Then, we evaluated the potency of spermidine to improve autophagy and mitophagy in our in vitro system.

## 2. Results

### 2.1. Spermidine Increases Bioenergetics and Cell Metabolic Activity in SH-SY5Y Cells 

To first examine the effect of spermidine in physiological (healthy) conditions, a screening was conducted on SH-SY5Y cells using the cell viability (MTT) assay and the ATP production assay, which are indicators of mitochondria metabolic activity. Five concentrations ranging from 0.05 μM to 10 μM were tested, as well as two treatment durations: 24 h and 48 h. No effects of spermidine were observed after 24 h of treatment (Figure 1A,B). However, 0.1 and 1 μM spermidine significantly increased cell metabolic activity and ATP production after 48 h of treatment (Figure 1C,D). The most efficient spermidine concentration was 0.1 μM, which induced an 8.2% increase in cell metabolic activity and a 4% increase in ATP production. The concentration of 0.1 μM of spermidine and 48 h of treatment duration were therefore selected to conduct further investigations in a cellular model of tauopathy. 

### 2.2. Spermidine Increases Bioenergetics in P301L Tau-Expressing Cells

As already shown in former studies [9,10,11], P301L tau-expressing cells (P301L cells) present impairments in mitochondrial bioenergetics and an increase in ROS levels. Indeed, P301L cells exhibit a decrease in cell metabolic activity (Figure 2A) in ATP concentration (Figure 2B) and MMP (Figure 2C), as well as an increase in superoxide anion levels (total and mitochondrial, Figure 2D,E) when compared with control (vector) cells. Based on our pre-screening data obtained in healthy cells (native cells, Figure 1), we assessed whether treatment with 0.1 μM spermidine for 48 h could improve bioenergetic parameters not only in the control cells but also in P301L cells (Figure 2). 

Cell metabolic activity was significantly increased in both vector and P301L after spermidine treatment (+14% and +24%, respectively, Figure 2A). Of note is that the cell metabolic activity of P301L cells was restored to the level of vector cells after spermidine treatment. A 48 h treatment with 0.1 μM spermidine also significantly increased the ATP level in vector cells (+6%) and in P301L cells (+5%) compared with untreated cells (Figure 2B). Similarly, spermidine increased the MMP in vector cells (+18%), as well as in P301L cells with a higher efficacy (+29%) compared with the untreated groups (Figure 2C).

Regarding the effects of spermidine on ROS levels, a 48 h treatment significantly reduced total and mitochondrial superoxide anions (O2•(−)) level in P301L cells (−5% and −7%, respectively), but not in vector cells (Figure 2D,E).

### 2.3. Spermidine Increases Mitochondrial Respiration and Cellular Glycolysis in P301L Tau-Expressing Cells

We next assessed whether spermidine could improve the bioenergetic profile of mutant tau cells. Indeed, in line with previous studies [10], P301L tau-expressing cells presented a decrease in oxygen consumption rate (OCR), an indicator of mitochondrial respiration (Figure 3A).

A 48 h treatment with 0.1 μM spermidine improved basal mitochondrial respiration in both control (vector) and P301L cells (Figure 3A,B). Other respiratory parameters were also calculated from the OCR values after the injection of specific inhibitors (Figure 3A,B). Besides the basal respiration, spermidine significantly increased the maximal respiration, the spare respiratory capacity and non-mitochondrial respiration in vector cells (Figure 3B). In P301L cells, spermidine increased ATP-linked respiration and maximal respiration when compared with untreated cells (Figure 3B). Additionally, spermidine also increased in the basal glycolysis (extracellular acidification rate, ECAR) in P301L cells compared with untreated cells (Figure 3C,D). 

The bioenergetic profile of vector and P301L cells, representing OCR versus ECAR (Figure 3C), revealed that cells were switched to a metabolically more active state, with a tendency to increase both glycolytic activity (ECAR) and basal respiration (OCR) after spermidine treatment.

Overall, these data indicate that spermidine is not only able to improve bioenergetic parameters in control cells (vector), but also to rescue some of these parameters in P301L cells. 

### 2.4. Spermidine Attenuates Impaired Autophagic Flux in P301L Tau-Expressing Cells

To investigate the impact of P301L tau on autophagy, vector and P301L, cells were transfected with LC3-DsRed (microtubule-associated protein 1A/1B-light chain 3-DsRed) and the percentage of LC3 puncta per cell area was measured (Figure 4A). Consistently with previous studies [21], we observed that the number of LC3 puncta were significantly higher in P301L cells compared with vector cells (Figure 4B, basal). As increased LC3 puncta can either indicate enhanced autophagosome formation or impaired autophagosome degradation, we further investigated the role of P301L tau on autophagic flux using the autophagosome–lysosome fusion inhibitor bafilomycin A1 (BAFA1). Cells were treated with spermidine (0.1 μM) for 48 h, followed by BAFA1 100 nM or the vehicle dimethylsulfoxyde (DMSO) alone (0.01%). Following BAFA1 treatment, no significant difference in LC3 levels was observed between P301L cells and vector cells treated with vehicle (*p* > 0.9999), suggesting that elevated LC3 levels observed in P301L cells at baseline were due to the impaired degradation of autophagosomes, since equal levels of autophagosome formation were observed in both P301L cells and vector cells following BAFA1 treatment (Figure 4B, +BAFA1).

Intriguingly, spermidine treatment appeared to enhance both autophagosome formation and degradation. Treatment with spermidine significantly attenuated LC3 levels in P301L cells at baseline, indicating the enhanced degradation of autophagosomes. Simultaneously, spermidine treatment significantly increased LC3 levels in both P301L cells and vector cells following BAFA1 treatment, indicating enhanced autophagosome formation.

### 2.5. Spermidine Attenuates Mitophagy Deficits in P301L Tau Cells

To measure mitophagy, vector and P301L cells were transfected with LC3-DsRed and then stained with Mitotracker to assess the colocalization of LC3 puncta with mitochondria (Figure 5A). 

Cells were then stimulated with either vehicle (baseline, DMSO, <0.01%) or the mitochondrial uncoupler FCCP (carbonyl cyanide p-trifluoro-methoxyphenyl hydrazone, 10 μM) overnight to induce mitophagy. At baseline, no significant effect of either P301L expression or spermidine treatment on mitophagy levels was observed (Figure 5B, all *p* > 0.9816). However, following FCCP stimulation, P301L cells exhibited significantly reduced mitophagy levels compared with control cells. Treatment with spermidine significantly attenuated mutant tau-induced mitophagy impairments in P301L cells compared with treatment with vehicle alone following FCCP stimulation, returning mitophagy levels to those of vector controls. Similarly, a trend towards increased mitophagy in FCCP-treated vector cells treated with spermidine compared with vehicle alone was observed, but this did not reach significance (*p* = 0.5771).

To corroborate these findings, gene expression levels of proteins involved in the autophagy and mitophagy pathways were measured by RT-qPCR (Figure 5C–F). At baseline, no significant effect of P301L expression on the mitophagy genes *PARK2* (parkin) and *PINK1* (PTEN-induced putative kinase 1)was observed compared with control cells (Figure 5C,D). However, following FCCP stimulation, *PARK2* was significantly decreased in P301L tau-expressing cells compared with controls. Treatment with spermidine significantly increased *PARK2* gene expression levels in both P301L and vector cells following FCCP treatment (Figure 5D). Treatment with spermidine also increased *PARK2* expression at baseline in both cell lines. *PINK1* expression was only slightly decreased in P301L cells after FCCP stimulation (*p* = 0.0535) and no effect of spermidine treatment was observed at baseline or following FCCP stimulation. Decreased *LC3* levels were also observed in P301L cells compared with healthy controls following FCCP stimulation (Figure 5E). Spermidine treatment significantly elevated *LC3* levels in P301L cells at baseline, and a similar effect was observed in vector cells. Conversely, *P62* (sequestosome-1) expression was significantly elevated in P301L cells compared with control cells in both baseline conditions and following FCCP stimulation (Figure 5F). Intriguingly, spermidine treatment significantly increased *P62* expression in both P301L cells, as well as vector cells following FCCP stimulation compared with vehicle treatment. 

## 3. Discussion

In this study, we hypothesized that spermidine attenuates the detrimental effects of disease-associated tau on mitochondrial function. We confirmed previous works, showing that P301L tau-expressing cells not only exhibit impaired mitochondrial bioenergetics, but also impaired mitophagy [9,10,11,12]. Notably, we demonstrated that a 48 h treatment with spermidine improved bioenergetics and autophagy/mitophagy in the presence of P301L mutant tau.

Our findings are in line with previous work performed on neuroblastoma cells, which showed that spermidine improved mitochondrial function in an in vitro model of aging [20]. In this study, N2a cells were treated with d-galactose (d-Gal) to establish cell aging, and the anti-aging effects of a pre-treatment with spermidine were investigated. Treatment with spermidine delayed cell aging, ameliorated ATP production, increased oxygen consumption and maintained MMP. Spermidine also enhanced autophagy after d-Gal treatment. In addition, spermidine has previously been shown to increase autophagy in vitro in different cell lines, but also in vivo in different tissues (reviewed in [22]). Namely, spermidine increased LC3 expression in d-Gal-treated N2a cells [20]. Similarly, LC3 protein level was increased in brain tissues of senescence-accelerated mouse prone 8 (SAMP8) mice after 8 weeks of treatment (spermidine 2 mM in drinking water) [23]. Spermidine also increased other autophagy-related proteins such as Beclin 1 and P62. In line with these findings, in the same mouse model, spermidine increased the level of mitochondrial fusion proteins, mitofusin one and two and decreased the level of mitochondrial fission protein, dynamin-related protein one, paralleled with an increase in cytochrome c oxidase (COX IV) and ATP concentration [24]. Other beneficial effects of spermidine were observed in this mouse model of aging, including decreased apoptosis and inflammation in the brain, increased neurotrophic factors, including nerve growth factor (NGF), PSD95 postsynaptic density proteins PSD95 and PSD93 and brain-derived neurotrophic factor (BDNF) in neurons, as well as improved cognitive function when compared to untreated SAMP8 mice. 

Schroeder and colleagues recently showed that a supplementation of dietary spermidine improved cognitive functions in aged mice and flies, and correlated with cognitive performance in humans [23]. In this study, they showed that spermidine passes the blood–brain barrier (BBB), as the polyamine was detected after 1 week in the brain of aged mice (18 month old) fed with spermidine via drinking water. Dietary spermidine also improved mitochondrial respiration and cognitive function in both aged mice and flies, and these effects were linked to the spermidine-mediated modulation of the autophagy/mitophagy pathways, especially the PINK1-dependent quality control pathway. Our findings are in line with these studies, but add new evidence showing that spermidine increases the expression of *PARK2* and *LC3* in basal conditions in both vector and P301L cells, as well as increases the expression PARK2 and *P62* in FCCP-treated cells (Figure 5).

Mitochondrial dysfunction is not only a characteristic of aging, but it is a hallmark of diseases such as AD and tau-related neurodegenerative disorders [3,7,25,26]. Besides, impaired autophagy has recently been identified as an important feature contributing to AD progression [27]. 

Dietary supplementation with spermidine (3 mM in drinking water for up to 290 days) was recently shown to decrease toxic soluble amyloid-β (Aβ) levels, as well as neuroinflammation in a mouse model of AD-related amyloidopathy (APPPS1 mice) [28]. Spermidine mostly acted on microglial cells and upregulated the autophagy pathway, leading to an increase in Aβ clearance. In line with this, in N2a cells overexpressing the amyloid precursor protein (APP), spermidine increased autophagic flux and LC3 levels, enhancing the clearance of APP clusters [29].

To our knowledge, no previous studies have investigated the effect of spermidine on abnormal tau-induced autophagy and mitochondrial dysfunction. Disruption of polyamine homeostasis has been observed in a mouse model of tauopathy (rTg4510 mice bearing the P301L mutation), specifically upregulated spermidine synthase, as well as increased acetylspermidine (AcSPD) levels [30]. Of note is that, while spermidine prevented tau fibrillization, AcSPD increased tau fibrillization and promoted tau oligomerization, suggesting different impacts of polyamines versus acetylated polyamines on tau biology. Increasing spermidine level might therefore prevent tau pathology. 

Our study is the first to assess the effects of spermidine on autophagy/mitophagy and mitochondrial dysfunction in a cellular model of tauopathy. Namely, we showed that spermidine improved mitochondrial respiration (OXPHOS), mitochondrial membrane potential, as well as ATP production in P301L tau-expressing cells (Figure 6). We also showed that spermidine significantly decreased free radical levels only in P301L cells, which exhibited higher levels of ROS compared with vector cells in basal condition. Additional experiments are needed to assess the effects of spermidine on the antioxidant system (glutathione system, superoxide dismutase expression/activity), as the increase in free-radical levels is a hallmark of brain aging and neurodegenerative disorders. Studies in which oxidative damages are induced (e.g., using hydrogen peroxide) would help elucidate the effect of spermidine on oxidative stress in healthy cells. In the present study, spermidine increased autophagosome formation and degradation, upregulated the expression of *LC3*, *P6*2 and *Parkin* and restored P301L tau-induced impairments in mitophagy. Vector control cells treated with spermidine also exhibited significantly increased *P62* and *Parkin* expression, and a trend towards increased mitophagy following FCCP treatment was observed; however, this did not reach significance. This may be due to a ceiling effect, in which severe perturbations in mitochondrial status caused by FCCP induce maximal mitophagy capacity, rendering it difficult to detect further increases in healthy cells. Future experiments using more gentle stimuli with less mitochondrial perturbations, such as antimycin A or valinomycin, may help elucidate the effect of spermidine on mitophagy in healthy cells.

In humans, the main sources of spermidine comes from the intake of polyamine-rich foods, microbial synthesis by gut bacteria and cellular synthesis [31]. Among plant-derived foods, spermidine is found in particularly high levels in wheat germ and soybeans. Spermidine is also found in significant amounts in mushrooms, peas, hazelnuts, pistachios, spinach, broccoli, cauliflower and green beans. In animal-derived foods, meat and its derivatives show the highest polyamine contents, especially spermidine. Polyamines are also found in moderate quantities in fish and its derivatives, while milk and eggs only contain low amounts [32].

Interestingly, a recent study highlighted a correlation between oral spermidine intake and improved cognitive performance in subjects with mild and moderate dementia [33]. This study suggests that nutritional intervention with oral spermidine supplementation could prevent cognitive deficits in the early stages of dementia. Therefore, spermidine-enriched food might represent an attractive therapeutic approach to prevent/delay AD-related impairments. Our study corroborates these findings, as we demonstrated that spermidine improves neuronal bioenergetics and autophagy/mitophagy in the presence of disease-associated tau protein.

Further studies are now needed to confirm these findings, first on other cellular models, then in vivo. Indeed, we focused here on P301L tau because this mutation induces NFTs in mice that are similar to those observed in the brains of AD patients. The effects of spermidine should also be evaluated in the presence of other tau mutations (e.g., R406W, E10+16) involved in other tauopathies. Additionally, here we used SH-SY5Y cells that are human neuron-like cells. Studies on more advanced cellular models such as human induced-pluripotent stem cells (hiPSCs)-derived neurons would be more relevant to assess the effects of spermidine on human brain physiology.

## 4. Materials and Methods

### 4.1. Chemicals and Reagents

Dulbecco’s modified Eagle medium (DMEM), phosphate-buffered saline (PBS), fetal calf serum (FCS), Hanks’ Balanced Salt solution (HBSS), penicillin/streptomycin, dihydroethdium (DHE), thiazolyl blue tetrazolium bromide (MTT) and spermidine (#85558) were purchased from Sigma-Aldrich (St. Louis, MO, USA). MitoSOX and glutaMax were from Gibco Invitrogen (Waltham, MA, USA). Tetramethylrhodamine, methyl ester, perchlorate (TMRM) and MitoTrackerRed CMXROS were from Thermo Fisher Scientific (Waltham, MA, USA). The ATPlite1step kit was from PerkinElmer (Waltham, MA, USA) and horse serum (HS) was from Amimed, Bioconcept (Allschwil, Switzerland). Seahorse XFp Cell Mito Stress Test Kit, Seahorse XF Calibrant Solution, Seahorse XF DMEM Assay Medium, pH 7.4, glucose, pyruvate and glutamine were obtained from Agilent Technologies (Santa Clara, CA, USA). The RNA extraction kit was from Qiagen (Hilden, Germany); the GoScript™ Reverse Transcription Mix, Oligo and the GoTaq^®^ Master Mix for real-time quantitative PCR (RT-qPCR) were from Promega (Dübendorf, Switzerland). The blasticidin was from InvivoGen (San Diego, CA, USA).

### 4.2. Cell Culture

Human neuroblastoma SH-SY5Y cells (ATCC^®^ CRL-2266™ Manassas, VA, USA) are a well-established and widely used neuronal model in biochemical studies in general, as they express neuronal receptors. P301L-expressing SH-SY5Y human neuroblastoma cells were kindly provided by the laboratory of Jürgen Götz (Queensland Brain Institute, University of Queensland, Brisbane, Australia), and were generated using lentiviral gene transfer [34,35]. A concentration of 4.5 μg/mL blasticidin was added to the culture medium to select cell clones stably expressing the full-length human hTau40 bearing the P301L mutation and a green fluorescent protein (GFP) tag, or cells expressing the GFP-vector only (vector cells). Of note is that these specific cell lines stably expressing P301L tau (P301L cells) and the empty plasmid (vector cells) were used to avoid artefacts due to transient protein expression, which can be a cellular stress affecting mitochondrial physiology. Cells were grown and maintained in DMEM (D6429) containing glucose (4.5 g/mL), L-glutamine (0.584 g/L), sodium pyruvate (0.11 g/L), sodium bicarbonate (3.7 g/L) and phenol-red, and were supplemented with 10% heat-inactivated fetal calf serum (FCS), 5% horse serum (HS), 1% penicillin-streptomycin and 1% Glutamax at 37 °C in 5% CO_2_. The cells were kept in culture in 10 cm^2^ dishes, split twice a week and plated when they reached around 80% confluence, 1 day prior to treatment. 

### 4.3. Treatment Paradigm

Cells were seeded in different cell plates depending on the parameter to assess, and treated with spermidine after 24 h post-seeding. Spermidine was prepared from a stock solution at 10 mM in water. Pre-screening experiments were performed on control cells to determine the best spermidine concentration and treatment duration. ATP and MTT assays were used as readouts (Figure 1). Based on the data obtained, a 48 h treatment with 0.1 μM spermidine was subsequently used to assess different bioenergetic parameters, as well as the effects of spermidine on autophagy/mitophagy in vector versus P301L cells. Of note is that, since vector and P301L cells are tagged with GFP, the proliferative effect of spermidine was assessed by measuring the GFP fluorescent signal. No significant differences were observed between untreated and spermidine-treated conditions in both cell lines (*p* = 0.638), indicating that spermidine does not affect cell proliferation. 

### 4.4. Cell Viability Assay

Cell viability was investigated using an MTT assay. SH-SY5Y cells were plated in at least 5 replicates into 96-well cell culture plates at a density of 1.5 × 10^4^ cells/well. After spermidine treatment, the cells were incubated with 5 mg/mL MTT (3-(4,5-dimethylthyazol-2-yl)-2,5-diphenyl-tetrazolium bromide) in DMEM (10 µL/well) for 2 h. MTT is reduced to a violet formazan derivative by mitochondrial enzymatic activity. 

Subsequently, the medium was removed and 200 µL of DMSO was added to each well to dissolve the formazan crystals. MTT absorbance was measured at 550 nm using the multiplate reader Cytation 3 (BioTek, Luzern, Switzerland).

### 4.5. ATP Levels

Total ATP content was determined using a bioluminescence assay (ATPlite 1step, Perkin Elmer) according to the manufacturer’s instructions. Briefly, SH-SY5Y cells were plated in at least 5 replicates into white 96-well cell culture plates at a density of 1.5 × 10^4^ cells/well. The method measures the formation of light from ATP and luciferin catalyzed by the enzyme luciferase. The emitted light was linearly correlated to the ATP concentration and was measured using the multiplate reader Cytation 3 (BioTek).

### 4.6. Determination of Mitochondrial Membrane Potential (MMP)

The MMP was measured using fluorescent dye tetramethylrhodamine, methyl ester and perchlorate (TMRM). Cells were plated in at least 5 replicates into a black 96-well cell culture plate at a density of 1.5 × 10^4^ cells/well. Cells were loaded with the dye at a concentration of 0.4 μM for 20 min. After washing twice with 200 μL HBSS, the fluorescence was detected using the multiplate reader Cytation 3 (BioTek) at 530 nm (excitation)/590 nm (emission). The fluorescence intensity of the dye is dependent on the MMP.

### 4.7. Determination of Superoxide Anion Radical Levels

Total and mitochondrial superoxide anion radical levels were assessed using dihydroethidium (DHE) and red mitochondrial superoxide indicator (MitoSOX), respectively. Cells were plated in at least 5 replicates into black 96-well cell culture plates at a density of 1.5 × 10^4^ cells/well. After treatment, cells were incubated with 10 μM of DHE for 20 min or with 5 μM of MitoSOX for 90 min at room temperature in the dark on an orbital shaker. After washing the cells three times with HBSS, the formation of red fluorescent products were detected at 531 nm (excitation)/595 nm (emission). The intensity of fluorescence was proportional to the total and mitochondrial superoxide anion levels. The fluorescence was measured using the multiplate reader Cytation 3 (BioTek).

### 4.8. Oxygen Consumption Rate and Extracellular Acidification Rate

Key parameters related to mitochondrial respiration were investigated using the Seahorse XF HS Mini Analyzer (Agilent), allowing for simultaneous real-time measurement of the oxygen consumption rate (OCR) and the extracellular acidification rate (ECAR). Cells were plated with 3 replicates into a Seahorse XFp Cell Culture Miniplate (Agilent Technologies) at a density of 1.5 × 10^4^ cells per well. The following day, the XF Mito Stress Test protocol was performed according to the manufacturer’s instructions. For the measurement, the assay medium consisted of the Seahorse XF DMEM medium, pH 7.4 (Agilent Technologies) supplemented with 18 mM glucose, 4 mM pyruvate and 2 mM L-glutamine. The OCR and ECAR were recorded simultaneously, first under basal conditions, followed by the sequential injection of oligomycin (1.5 µM), carbonyl cyanide-p-trifluoromethoxyphenylhydrazone (FCCP, 1 µM) and a combination of antimycin A (0.5 µM) and rotenone (1 µM). The obtained data were analyzed on the Agilent Seahorse Analytics website, which automatically calculated the bioenergetic parameters, including basal respiration, proton leak, maximal respiration, spare respiratory capacity, non-mitochondrial oxygen consumption and ATP-production coupled respiration.

### 4.9. Assessment of Mitophagy/Autophagy

Cells were plated in 12-well plates containing coverslips coated with collagen (0.1 mg/mL; Sigma) at a density of 5 × 10^5^ cells per well. For experiments investigating autophagy and mitophagy, cells were transiently transfected with pmRFP-LC3 (Addgene #21075) using the Xfect™ Transfection Reagent (Takara Bio # 631317) according to the manufacturer’s recommendation. Twenty-four hours after transfection, cells were incubated with either 0.1 μM of spermidine or vehicle (water) for 48 h. To assess autophagic flux, cells were stimulated with 100 nM bafilomycin A1 or DMSO (vehicle) for 4 h prior to fixation with 4% paraformaldehyde. To assess mitophagy, cells were stimulated with 10 μM carbonyl cyanide p-trifluoro-methoxyphenyl hydrazone (FCCP) overnight, followed by incubation with 100 nM Mitotracker (Thermo Fisher, M22426) for 40 min rotating in the dark. Following the indicated treatment, the cells were fixed with 4% paraformaldehyde, mounted on slides, and stored for imaging.

### 4.10. Microscopy and Image Analysis

Images were captured using an inverted microscope (Leica Microsystems TCS SPE DMI4000, Wetzlar, Germany) attached to an external light source for enhanced fluorescence imaging (Leica EL6000) with Leica LAS AF imaging software. Each dataset was imaged in a single session, with the same imaging settings maintained throughout the session. Image analyses were performed in a user-blinded manner using ImageJ and de-noised using a background subtraction rolling ball radius of 50 pixels. For display, images were adjusted for brightness and contrast, with channel minimum and maximum values kept consistent between images. Maximum projections were used for analysis.

To quantify autophagy, high-resolution z-stacks imaged at 40× resolution with an optimal step size were performed to capture multiple cells across the coverslip. Regions of interest (ROIs) were drawn around single cells in ImageJ and selected in the 568 nm channel containing the LC3 signal. The fluorescence intensity of LC3-RFP was measured above a set threshold to exclude background pixel values. To quantify mitophagy, both LC3-RFP and Mitotracker were measured above a set threshold to exclude background pixel values, and the percentage of LC3 puncta that colocalized with mitochondria was calculated for each cell.

### 4.11. RNA Extraction and Real-Time Quantitative PCR

For quantitative PCR (qPCR), total RNA was isolated from cells using the RNeasy Mini kit (Qiagen, 74104), as per the manufacturer’s instructions. cDNA synthesis was carried out using the GoScript™ Reverse Transcription Kit (Promega, A2791) in an RNase-free environment. Only RNA samples with 260/280 nm of 2.0 ± 0.2 were used. Real-time PCR, executed with the GoTaq**^®^** qPCR kit (Promega, A6002), was used to amplify the standards and then quantify the sample’s mRNA expression. Using 96-well PCR plates (Thermo Fisher, AB0800W) covered by adhesive seals (4titutde, 4ti-0565), duplicates of each sample were mixed with SYBR green dye and appropriate primers. The sequence of the primers is mentioned in Table 1. Gene expression was measured as fold change and was evaluated by the 2^−ΔCT^ method. The data are represented as relative mRNA expression normalized to human GAPDH mRNA expression, and are relative to a reference sample containing the pool of all the samples [36].

## Figures and Tables

**Figure 1 ijms-24-05297-f001:**
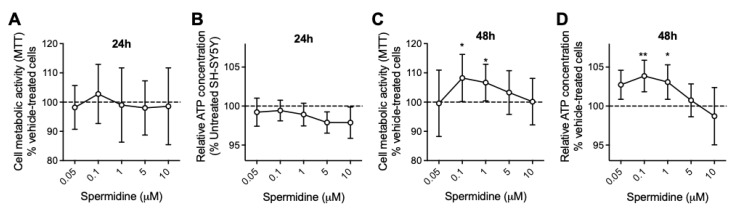
Effects of spermidine on cell metabolic activity and ATP production. (**A**) No effect of spermidine on cell metabolic activity or (**B**) ATP production was observed after 24 h of treatment. (**C**) Spermidine at 0.1 and 1 μM significantly increased the metabolic activity in SH-SY5Y cells after 48 h of treatment. (**D**) Spermidine at 0.1 and 1 μM significantly increased ATP production in SH-SY5Y cells after 24 h of treatment. Values represent the mean and SD of four independent experiments (N = 20–25 replicates per condition). * *p* < 0.05; ** *p* < 0.01; one-way ANOVA and post hoc Dunnett’s multiple comparison test versus control condition (vehicle-treated cells). MTT: MTT assay.

**Figure 2 ijms-24-05297-f002:**
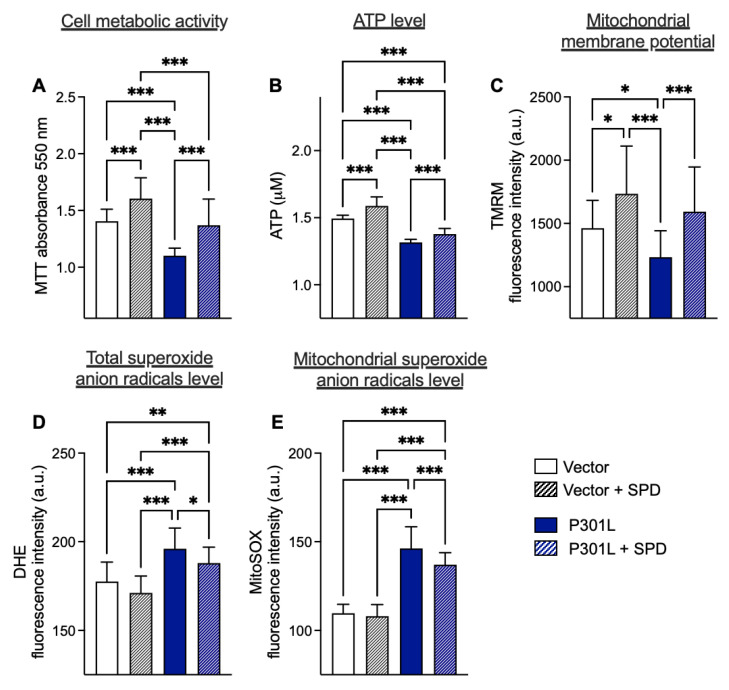
Improved mitochondrial bioenergetics and decreased reactive oxygen species levels by spermidine in control and P301L cells. Cells were treated with 0.1 μM spermidine for 48 h. (**A**) Cell metabolic activity in vector and P301L cells. (**B**) ATP concentration in vector and P301L cells. (**C**) Mitochondrial membrane potential in vector and P301L cells. (**D**) Total superoxide anion level in vector and P301L cells. (**E**) Mitochondrial superoxide anion level in vector and P301L cells. Data are presented as mean ± SD (N = 25–30 replicates/conditions). * *p* < 0.05; ** *p* < 0.01; *** *p* < 0.001; one-way ANOVA and post hoc Tukey’s multiple comparisons test. MTT: MTT assay, SPD: spermidine, TMRM: tetramethylrhodamine, methyl ester and perchlorate, DHE: dihydroethdium.

**Figure 3 ijms-24-05297-f003:**
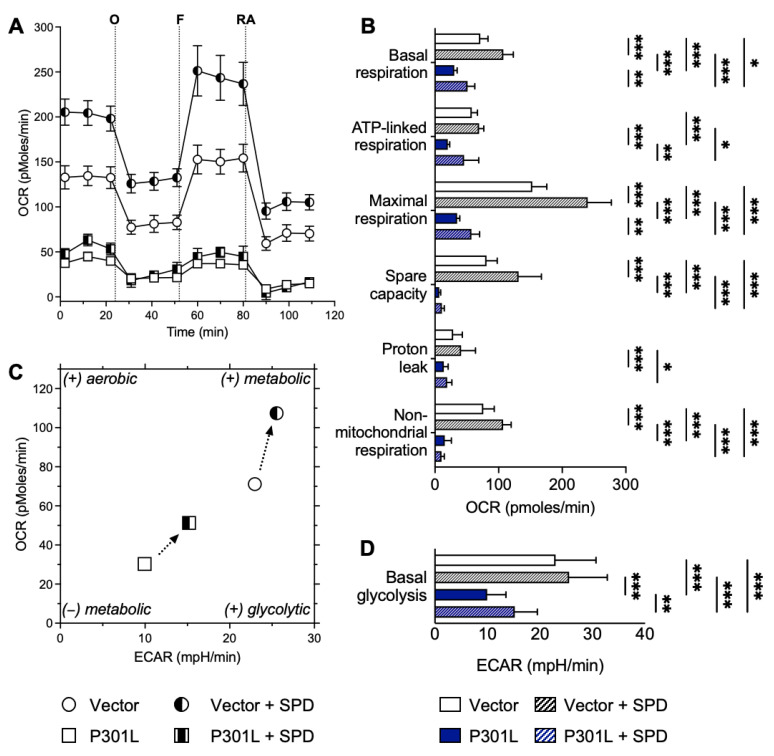
Effects of spermidine on the bioenergetic phenotype of P301L tau-expressing cells and control cells. (**A**) Oxygen consumption rate (OCR) of vector and P301L tau cells was measured using a Seahorse XF HS Mini Analyzer (Agilent). The sequential injection of mitochondrial inhibitors, namely oligomycin (O), FCCP (F) and rotenone/antimycin A (RA), is indicated (see details in the “Materials and Methods” section). Values represent the mean ± SEM. (**B**) Values corresponding to the different respiratory parameters of vector cells and P301L cells are represented as mean ± SD (n = 8–10 replicates/condition). * *p* < 0.05, ** *p* < 0.01, *** *p* < 0.001, two-way ANOVA + post hoc Tukey’s multiple comparisons test. (**C**) Bioenergetic phenotype (OCR versus ECAR) of vector cells and P301L cells revealed an increase in metabolic activity after a 48 h treatment with 0.1 μM spermidine. Values represent the mean of each group (mean of the ECAR in abscissa/mean of the OCR in ordinate). (**D**) Extracellular acidification rate (ECAR) corresponding to the basal glycolysis with or without spermidine (SPD) treatment are represented as mean ± SD (n = 8–10 replicates). ** *p* < 0.01, *** *p* < 0.001, one-way ANOVA + post hoc Tukey’s multiple comparisons test.

**Figure 4 ijms-24-05297-f004:**
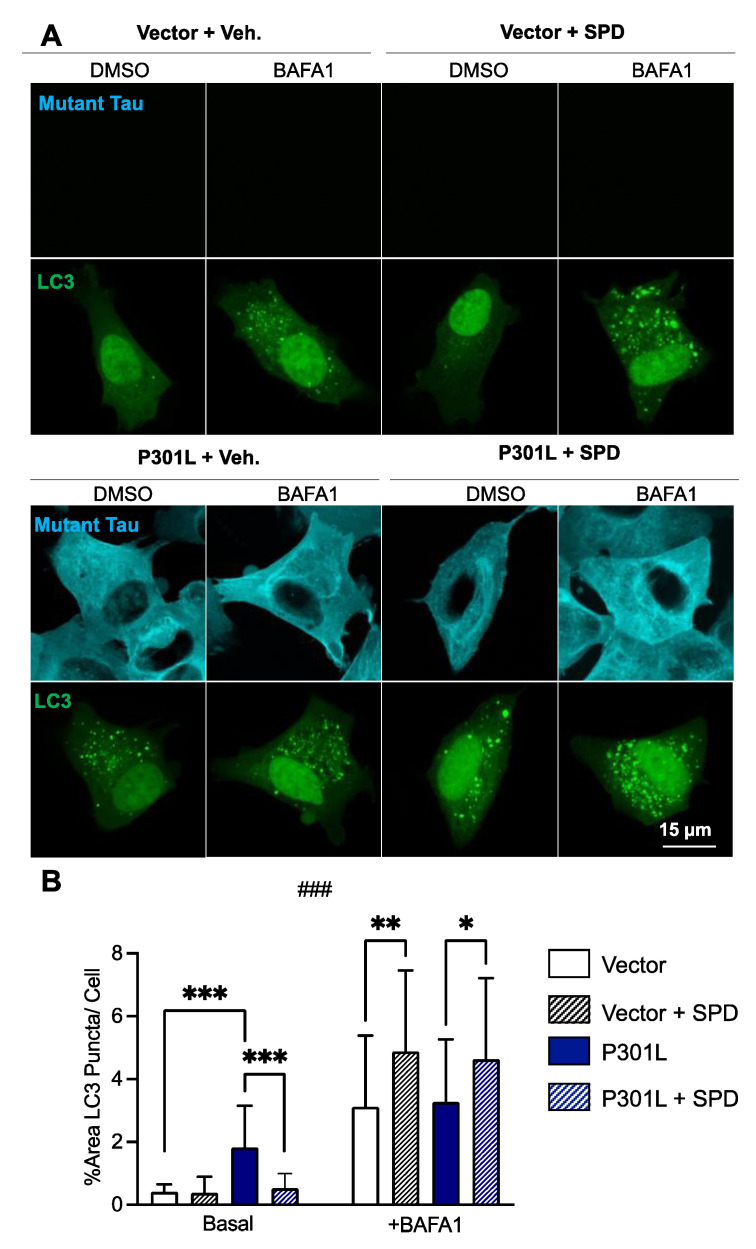
Effects of spermidine on autophagy in P301L tau-expressing cells and vector cells. (**A**) Autophagic flux was assessed in vector and P301L cells by stimulating them with the autophagosome–lysosome fusion inhibitor Bafilomycin A1 (BAFA1, 100 nM) or DMSO alone. LC3 puncta (green), indicators of autophagosome formation, were then quantified. (**B**) P301L tau-expressing cells exhibit increased LC3 puncta levels when compared with vector cells at baseline. Spermidine (SPD) attenuated increased LC3 levels in P301L cells in basal conditions and increased the autophagic flux following treatment with Bafilomycin A1 in both cell lines. Data are represented as mean ± SD (n = 30–50 cells/conditions). * *p* < 0.05, ** *p* < 0.01; *** *p* < 0.001; three-way ANOVA + post hoc Tukey’s multiple comparisons test, ### *p* < 0.001, three-way ANOVA baseline vs. BAFA1, LC3: microtubule-associated protein 1A/1B-light chain 3.

**Figure 5 ijms-24-05297-f005:**
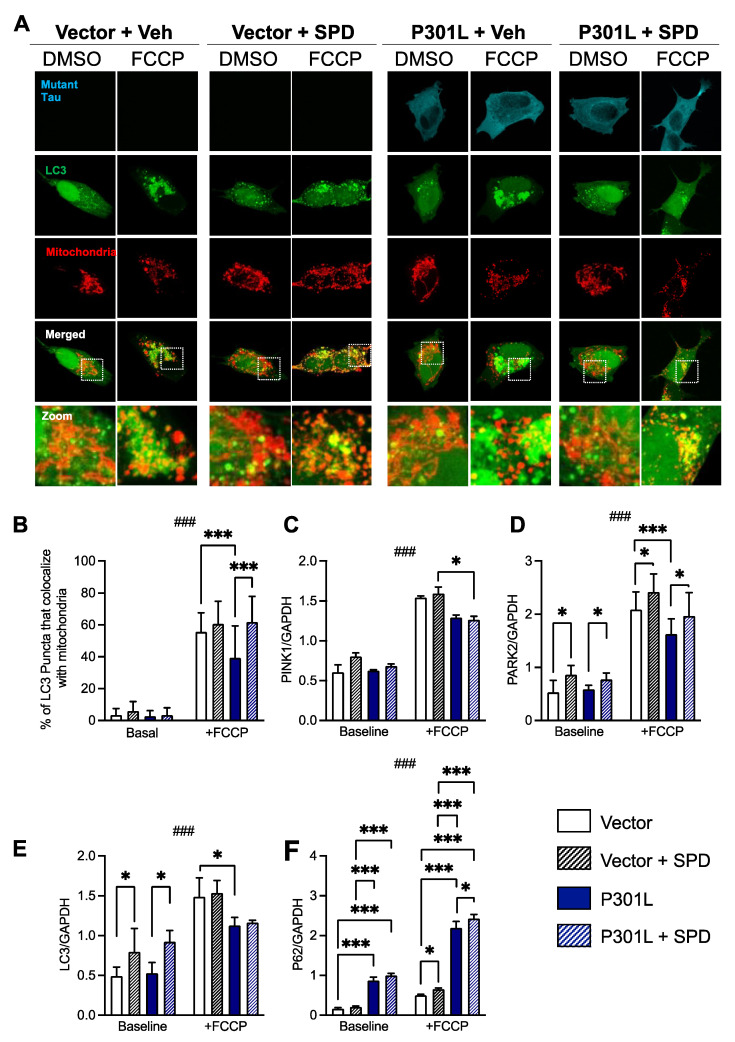
Effects of spermidine on mitophagy in P301L tau-expressing cells and control cells. (**A**) To assess the levels of basal and carbonyl cyanide p-trifluoro-methoxyphenyl hydrazone (FCCP)-induced mitophagy, P301L and vector cells were transfected with LC3-RFP (green) and then stained with mitotracker (red) to measure the colocalization of LC3 puncta with mitochondria. (**B**) In basal conditions, no significant effect of P301L tau on cellular mitophagy was observed. However, following stimulation with the protonophore FCCP, which induces mitophagy by collapsing the mitochondrial membrane potential, P301L cells exhibited significantly reduced mitophagy compared with controls. Spermidine (SPD) significantly increased mitophagy in P301L cells following FCCP treatment. (**C**–**F**) Impact of mutant tau and SPD on the mRNA level of proteins involved in mitophagy, namely PINK1 (**C**), PARK2 (**D**), LC3 (**E**) and P62 (**F**). * *p* < 0.05, *** *p* < 0.001; three-way ANOVA + post hoc Tukey’s multiple comparisons test, ### *p* < 0.001, three-way ANOVA baseline vs. FCCP. PINK1: PTEN-induced putative kinase 1; PARK2: Parkin; LC3: microtubule-associated protein 1A/1B-light chain 3; P62: Sequestosome-1.

**Figure 6 ijms-24-05297-f006:**
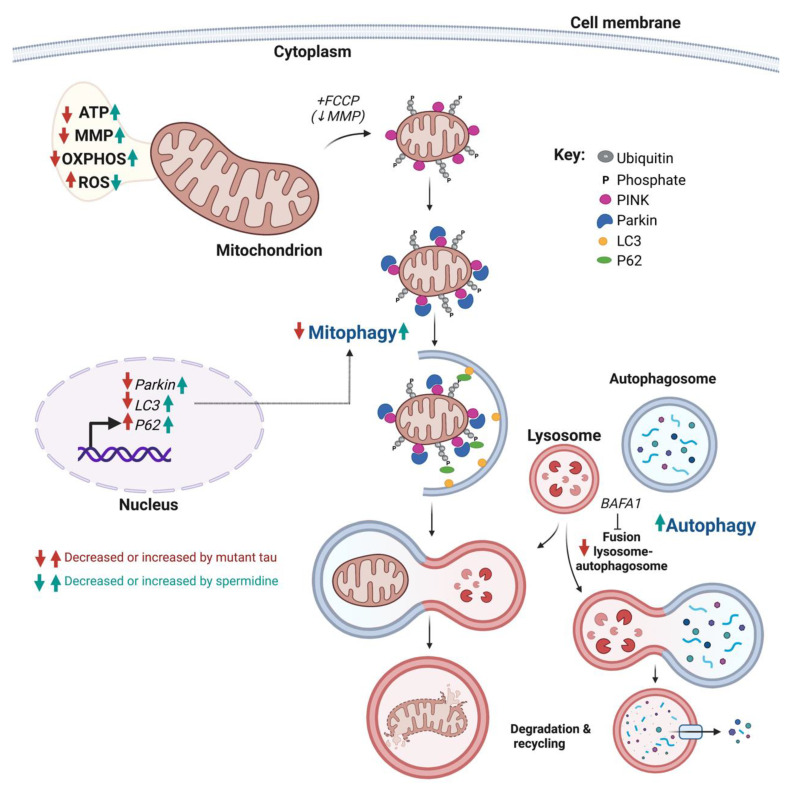
Schematic representation of the effects of abnormal tau protein and spermidine on mitochondrial function and autophagy. P301L mutant tau protein impairs mitochondrial bioenergetics, leading to decreased mitochondrial respiration, mitochondrial membrane potential (MMP) and ATP production, as well as an increase in reactive oxygen species (ROS) level. Spermidine seems to counteract the effects of abnormal tau on these parameters. P301L mutant tau protein also impairs autophagy (possibly by inhibiting the fusion between lysosomes and autophagosomes) as well as mitophagy. These effects may be, at least in part, due to the abnormal tau-induced dysregulation of genes involved in mitophagy/autophagy. Spermidine improves these processes in mutant tau-expressing cells and increases the expression of genes involved in mitophagy/autophagy. PINK1: PTEN-induced putative kinase 1; PARK2: Parkin; LC3: microtubule-associated protein 1A/1B-light chain 3; P62: Sequestosome-1. Created with BioRender.com.

**Table 1 ijms-24-05297-t001:** Primer sequences used for RTqPCR analysis.

Primer	Sequences
PARK2	F-5′-GGA AGT CCA GCA GGT AGA-3′ R-5′-ATC CCA GCA AGA TGG ACC-3′
PINK1	F-5′-CCA TCA AGA TGA TGT GGA ACA-3′ R-5′-GAC CTC TCT TGG ATT TTC TGT AA-3′
LC3	F-5′-CTC AGA CCG GCC TTT CAA-3′ R-5′-CGA TGA TCA CCG GGA TTT TG-3′
P62/SQSTM1	F-5′-CGG CAG AAT CAG CCT CTG-3′ R-5′-GTC AGG CGG CTT CTT TTC-3′
GAPDH	F-5′-CAT GGT TTA CAT GTT CCA ATA TGA-3′ R-5′-GGA TCT CGC TCC TGG AAG-3′

Abbreviations: PARK2, Parkin; PINK-1, PTEN-induced putative kinase 1; LC3, microtubule-associated protein light chain 3; P62/SQSTM1, Sequestosome-1; GAPDH, Glyceraldehyde 3-phosphate dehydrogenase; qRT-PCR, quantitative real-time polymerase chain reaction.

## Data Availability

The data presented in this study are available on request from the corresponding author.

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
