# Peer review of "Spermidine Rescues Bioenergetic and Mitophagy Deficits Induced by Disease-Associated Tau Protein"

_ijms, 2023, doi:10.3390/ijms24065297_

Round 1
Reviewer 1 Report
The article ‘Spermidine rescues bioenergetic and mitophagy deficits induced by disease-associated tau protein ' by Fairley et al. describes effects of SPD on abnormal tau-induced mitochondrial dysfunction. The study design and experimental approaches are usually appropriate. The authors claim that the SPD improves mitochondrial respiration, mitochondrial membrane potential as well as ATP production in both control and P301Ltau-expressing cells hence propose SPD supplementation as a therapeutic approach in tauopathy. The results presented here are informative and relevant for the 'IJMS.' The data presented here is clear and generally supports the conclusion. The authors need to address some concerns in the current version of the manuscript before its publication.
The Abstract is written in a very casual language. It needs to be rewritten.
In Figure 2 while the authors compare the effects of SPD in control and P301L separately, it would be more convincing and informative to compare the effects of SPD treatment on control and P301L in a single graph parallelly. The same is also recommended for Figure 3A and B. A consistent comparision has not been followed throughoutt he experiments. The authors compare some results in either control or P301L independently, and in other results (for example in 5B), the appropriate comparison is missing. The authors should discuss the absence of effects across treatments (and across control and experimental groups) in some experiments versus the others.
How do the authors explain no significant change in +FCCP in vector control after SPD treatment in Fig5B?
There are many verb tense inconsistencies throughout the manuscript. Further, the writing style, typographical and grammatical errors should be corrected in the revised version of the manuscript.
Author Response
The Abstract is written in a very casual language. It needs to be rewritten.
Answer : We thank the reviewer for the comment. The abstract has been rewritten. Please see lines 11-26: “Abnormal tau build-up is a hallmark of Alzheimer’s disease (AD) and more than 20 other serious neurodegenerative diseases. Mitochondria are paramount organelles playing a predominant role in cellular bioenergetics, namely by providing the main source of cellular energy via adenosine triphosphate generation. Abnormal tau impairs almost every aspects of mitochondrial function, from mitochondrial respiration to mitophagy. The aim of our study was to investigate the effects of spermidine, a polyamine which exerts neuroprotective effects, on mitochondrial function in a cellular model of tauopathy. Recent evidence identified autophagy as the main mechanism of action of spermidine on life-span prolongation and neuroprotection, but the effects of spermidine on abnormal tau-induced mitochondrial dysfunction have not yet been investigated. We used SH-SY5Y cells stably expressing a mutant form of human tau protein (P301Ltau mutation) or cells expressing the empty vector (control cells). We showed that spermidine improved mitochondrial respiration, mitochondrial membrane potential as well as ATP production in both control and P301Ltau-expressing cells. We also showed that spermidine decreased the level of free radicals, increased autophagy, and restored P301Ltau-induced impairments in mitophagy. Overall, our findings suggest that spermidine supplementation might represent an attractive therapeutic approach to prevent/counter-act tau-related mitochondrial impairments.”
In Figure 2 while the authors compare the effects of SPD in control and P301L separately, it would be more convincing and informative to compare the effects of SPD treatment on control and P301L in a single graph parallelly. The same is also recommended for Figure 3A and B. A consistent comparision has not been followed throughoutt the experiments. The authors compare some results in either control or P301L independently, and in other results (for example in 5B), the appropriate comparison is missing. The authors should discuss the absence of effects across treatments (and across control and experimental groups) in some experiments versus the others.
Answer : We thank the referee for this comments. The figures 2 and 3 were modified accordingly. We now compare every condition with each other (see new stats).
The absence of effects across treatments (and across control and experimental groups) in some experiments versus the others were discussed. Please see:
Lines 310-317: “We also showed that spermidine significantly decreased free radical levels only in P301L cells, which exhibited higher levels of ROS compared to vector cells in basal condition. Additional experiements are needed to assess the effects of spermidine on the antioxidant system (glutathione system, superoxide dismutase expression/activity), as the increase of free-radicals level is a hallmark of brain aging and neurodegenerative disorders. Studies in which oxidatve damages are induced (e.g. using hydrogen peroxide) would help elucidating the effect of spermidine on oxidative stress in healthy cells.”
Lines 320-327: “Vector control cells treated with spermidine also exhibited significantly increased P62 and Parkin expression, and a trend towards increased mitophagy following FCCP treatment was observed, however this did not reach significance. This may be due to a ceiling effect, in which severe perturbations in mitochondrial status caused by FCCP induce maximal mitophagy capacity, rendering it difficult to detect further increases in healthy cells. Future experiments using more gentle stimuli with less mitochondrial perturbations, such as antimycin A or valinomycin may help elucidate the effect of spermidine on mitophagy in healthy cells.”
How do the authors explain no significant change in +FCCP in vector control after SPD treatment in Fig5B?
Answer : We thank the referee for raising this point. We did observe a trend towards increased mitophagy in vector control cells treated with SPD, following FCCP stimulation, but this did not reach significance. The addition of FCCP is known to induce mitophagy in cells and we hypothesize that vector control cells treated with FCCP may already be initiating mitophagy processes to their maximum capacity, creating a ceiling effect. Thus, the addition of SPD is unlikely to further increase this process significantly. However, when a deficit in mitophagy capacity is present, such as in the P301L cells, SPD is able to rescue this deficit and stimulate mitophagy to levels comparable to the control cells.
We have edited the manuscript to better clarify this finding as follows in line 320-327:
“Vector control cells treated with spermidine also exhibited significantly increased P62 and Parkin expression, and a trend towards increased mitophagy following FCCP treatment was observed, however this did not reach significance. This may be due to a ceiling effect, in which severe perturbations in mitochondrial status caused by FCCP induce maximal mitophagy capacity, rendering it difficult to detect further increases in healthy cells. Future experiments using more gentle stimuli with less mtiochondrial pertubrations, such as antimycin A or valinomycin may help elucidate the effect of spermidine on mitophagy in healthy cells.”
There are many verb tense inconsistencies throughout the manuscript. Further, the writing style, typographical and grammatical errors should be corrected in the revised version of the manuscript.
Answer : We thank the reviewer for this comment. The manuscript has been proof-read by a native English speaker, who have edited the manuscript to remove any issues with the writing style, typographical and grammatical errors.
Reviewer 2 Report
This study tests an interesting idea, but I have a number of concerns related to data presentation, which might reflect issues underlying the experimental design. Otherwise, a better representation of the data throughout the manuscript, and without duplication, could solve the lack of clarity. Specific comments:
-How relevant is the choice of the cell model? Sushi cells hardly represent neurons in vitro or in the brain. This should be well explained in the manuscript.
- metabolic activity using MTT is not necessarily measuring metabolic rates. In fact, MTT increases might just represent cell proliferation; and it's reduction could mean cell death. A test to differentiate the 2 events would be good.
- The representation of ATP as fraction of a control experiment is limited in definitively supporting the authors interpretations. ATP should be shown in absolute concentrations, and even better would be showing ADP/ATP ratios since ATP alone tells us little bout the energy charge of the cell.
- Study design is poor in regards to data shown in figure 2. P301L effects can't be effectively ascertained. It appears that vector and P301L experiments were not done in parallel. But normalizing the spermidine groups to the respective control in the presence or absence of P301L, both at 100%, does not allow to pick up effects of P301L per se. Then P301L+spermidine is totaly biased. What is the dashed line for vector cells representing exactly? Is this what is presented in table 1?
- figure 5C on gene expression should be modified since it does not show data variance. I suggest replacing it by a scatter plot. With so few genes analyzed, there's no reason for not showing individual datasets to have an idea on variance. I suppose that this is exactly what is in table 2. I see no reason to show the same results twice.
- immunofluorescence microscopy experiments show cells with dotted mitochondria only. I am surprised that there is no elongated mitochondria in these cell line. This could be related to culturing conditions, which are poorly reported.
- Methodology needs rigorous reporting. A number of cell growing settings impacts metabolism, namely substrates in the medium (what's glucose concentration in the culturing medium? lipids? are there other substrates?)
- the source of reagents, biologicals (e.g. there's many DMEM variants, with diverse substrates and substrate concentrations, with phenolred that might interact with many spectrometry/fluorimetry assays, including MTT assay) and key materials needs to be reported in the manuscript.
-qPCR experiments: GAPDH is know to change expression in several metabolic insults. It needs to be demonstrated that it does not change in this setting. Normalization to more than one gene is preferable. The deltaCT method was used to normalize gene expression. Is there an experimental group used as reference of is the normalization done for all across-groups measurements as reference?
-The discussion misses a reflection on the limitations of the study
minor:
- the abstract is not clear on the reasoning to test spermidine in a tauopathy model.
- I suggest reporting all data as mean and SD, rather than using SEM.
- I suggest using SI units throughout the manuscript.
- I see no reason to abbreviate spermidine to SPD
-in table 1, measurements are relative to what?
Author Response
-How relevant is the choice of the cell model? Sushi cells hardly represent neurons in vitro or in the brain. This should be well explained in the manuscript.
Answer : We agree with the referee. SH-SY5Y is widely used neuronal model for in vitro study. The main advantage of using this model is that it is a human neuronal cell line, and, in our case, there were successfully stably transfected to overexpress P301L mutant tau protein. Indeed, getting a stably tau transfected cell line is very challenging, therefore this specific model is used to study the impact of abnormal tau on nerve cells physiology. Besides, the use of stably transfected cells avoids avoid artefacts due to a transient protein expression which can be a cellular stress affecting mitochondria physiology. A statement was added in the method part. Please see lines 390-395:
“P301L-expressing SH-SY5Y human neuroblastoma cells were kindly provided by the laboratory of Jürgen Götz (Queensland Brain Institute, University of Queensland, Brisbane, Australia), and were generated using lentiviral gene transfer [35,36]. To the culture medium 4.5 μg/ml blasticidin was added to select cell clones stably expressing the full-length human hTau40 bearing the P301L mutation and a green fluorescent protein (GFP) tag, or cells expressing the GFP-vector only (vector cells). Of note, this specific cells lines stably expressing P301L-tau (P301L cells) and the empty plasmid (Vector cells) were used to avoid artefacts due to transient protein expression which can be a cellular stress affecting mitochondrial physiology.”
- metabolic activity using MTT is not necessarily measuring metabolic rates. In fact, MTT increases might just represent cell proliferation; and it's reduction could mean cell death. A test to differentiate the 2 events would be good.
Answer: We thank the referee for this comment. MTT assay is widely used to screen the effects of drugs on the cell metabolic activity, proliferation and/or death. We agree that it does not directly allow to assess the metabolic rate in the cells. Therefore, complementary tests were performed, including the ATP assay, mitochondrial membrane potential measurement, mitochondrial respiration / glycolysis (Seahorse). In addition, we verified if spermidine had proliferative effects by measuring the GFP signal of the cells (because vector and P301L cells are GFP-tagged). Please see the figure in the attached pdf document.
A statement was also added in the method part. Please see: 407-412:
“Of note, since vector and P301L cells are tagged with GFP, the proliferative effect of spermidine was assessed by measuring the GFP fluorescent signal. No significant differences were observed between untreated and spermidine-treated conditions in both cell lines (p=0.638, data not shown), indicating that spermidine does not affect cell proliferation.”
- The representation of ATP as fraction of a control experiment is limited in definitively supporting the authors interpretations. ATP should be shown in absolute concentrations, and even better would be showing ADP/ATP ratios since ATP alone tells us little bout the energy charge of the cell.
Answer: We thank the reviewer for raising this point. We show now absolute ATP concentration in uM in Figure 2. Besides, data obtained with the Seahorse XF Analyzer allows to assess different parameters of the mitochondrial respiration (Figure 3). Namely, we show that spermidine increases the basal respiration in vector and P301L cells, and the ATP-linked respiration in P301L cells. Indeed, the decrease in oxygen consumption rate upon injection of oligomycin represents the portion of basal respiration that was being used to drive ATP production. This parameter indirectly shows ATP produced by the mitochondria that contributes to meeting the energetic needs of the cell(https://www.agilent.com/cs/library/usermanuals/public/XF_Cell_Mito_Stress_Test_Kit_User_Guide.pdf ).
- Study design is poor in regards to data shown in figure 2. P301L effects can't be effectively ascertained. It appears that vector and P301L experiments were not done in parallel. But normalizing the spermidine groups to the respective control in the presence or absence of P301L, both at 100%, does not allow to pick up effects of P301L per se. Then P301L+spermidine is totaly biased. What is the dashed line for vector cells representing exactly? Is this what is presented in table 1?
Answer : We agree with the referee. Experiments on P301L and vector cells were done in parallel, but our first idea was to highlight the effect of spermidine per se on both cell lines. However, to increase the clarity of our results, we now show the absolute values for each parameter (e.g. absorbance, ATP concentration, fluorescence intensity), as well as data of both cells lines on the same graphs. Please see the new Figure 2. Of note, we did the same (combining data) for the Figure 3.
- figure 5C on gene expression should be modified since it does not show data variance. I suggest replacing it by a scatter plot. With so few genes analyzed, there's no reason for not showing individual datasets to have an idea on variance. I suppose that this is exactly what is in table 2. I see no reason to show the same results twice.
Answer : We agree with the referee. We decided to show the individual graphs for each gene expression. This allows to better apprehend the effects of FCCP stimulation, the difference between the cell lines, as well as the effect of spermidine. Please see the new Figure 5.
- immunofluorescence microscopy experiments show cells with dotted mitochondria only. I am surprised that there is no elongated mitochondria in these cell line. This could be related to culturing conditions, which are poorly reported.
Answer: We thank the review for raising this point. FCCP is known to induce mitochondrial fragmentation (PMID: 20489733), resulting in the more punctate mitochondria observed in these images. However, in baseline conditions (without FCCP treatment), we did observe much more elongated mitochondria, consistent with what we would expect to see in these cell lines. We have included an enlarged version of Fig. 5A below, to better demonstrate this morphology, in the attached pdf document.
We have previously characterised the expression of mitochondrial fission/fusion proteins, and mitochondrial morphology in these cell lines. However, as this characterisation was not performed with spermidine or FCCP treatment, and has previously been reported elsewhere (https://doi.org/10.3390/ijms21176344), we felt it was not necessary to include the data in this manuscript.
- Methodology needs rigorous reporting. A number of cell growing settings impacts metabolism, namely substrates in the medium (what's glucose concentration in the culturing medium? lipids? are there other substrates?)
- the source of reagents, biologicals (e.g. there's many DMEM variants, with diverse substrates and substrate concentrations, with phenolred that might interact with many spectrometry/fluorimetry assays, including MTT assay) and key materials needs to be reported in the manuscript.
Answer: We thank the referee for making us aware of this point. More details were added in the Method part. Please see lines 367-422:
“Chemicals and Reagents:
Dulbecco’s-modified Eagle medium (DMEM, ), phosphate-buffered saline (PBS), fetal calf serum (FCS), Hanks’ Balanced Salt solution (HBSS), penicillin/streptomycin, dihydroethdium (DHE), thiazolyl blue tetrazolium bromide (MTT), and spermidine (#85558) were purchased from Sigma-Aldrich (St. Louis, MO, USA). MitoSOX and glutaMax were from Gibco Invitrogen (Waltham, MA, USA). Tetramethylrhodamine, methyl ester, perchlorate (TMRM) and MitoTrackerRed CMXROS were from Thermo Fisher Scientific (Waltham, MA, USA). ATPlite1step kit was from PerkinElmer (Waltham, MA, USA) and horse serum (HS) from Amimed, Bioconcept (Allschwil, Switzerland). Seahorse XFp Cell Mito Stress Test Kit, Seahorse XF Calibrant Solution, Seahorse XF DMEM Assay Medium, pH 7.4, glucose, pyruvate, and glutamine were obtained from Agilent Technologies (Santa Clara, CA, USA). The RNA extraction kit was from Qiagen (Hilden, Germany), the GoScript™ Reverse Transcription Mix, Oligo, and the GoTaq® Master Mix for real-time quantitative PCR (RT-qPCR) from Promega (Dübendorf, Switzerland). The blasticidin was from InvivoGen (San Diego, CA, USA).
Cell Culture
Human neuroblastoma SH-SY5Y cells (ATCC® CRL-2266™ Manassas, VA, USA) are a well-established and widely used neuronal model in biochemical studies in general as it expresses neuronal receptors. P301L-expressing SH-SY5Y human neuroblastoma cells were kindly provided by the laboratory of Jürgen Götz (Queensland Brain Institute, University of Queensland, Brisbane, Australia), and were generated using lentiviral gene transfer [35,36]. To the culture medium 4.5 μg/ml blasticidin was added to select cell clones stably expressing the full-length human hTau40 bearing the P301L mutation and a green fluorescent protein (GFP) tag, or cells expressing the GFP-vector only (vector cells). Of note, this specific cells lines stably expressing P301L-tau (P301L cells) and the empty plasmid (Vector cells) were used to avoid artefacts due to transient protein expression which can be a cellular stress affecting mitochondrial physiology. Cells were grown and maintained in DMEM (D6429) containing glucose (4.5 g/mL), L-glutamine (0.584 g/L), sodium pyruvate (0.11 g/L), sodium bocarbonate (3.7 g/L), phenol-red, and supplemented with 10% heat‐inactivated fetal calf serum (FCS), 5% horse serum (HS), 1% penicillin-streptomycin, and 1% Glutamax at 37°C in 5% CO2. The cells were kept in culture in 10 cm2 dishes, split twice a week and plated when they reached around 80% confluence, 1 day prior treatment.
Treatment Paradigm
Cells were seeded in different cell plates, depending on the parameter to assess, and treated with spermidine after 24 hrs post-seeding. Spermidine was prepared from a stock solution at 10 mM in water. Pre-screening experiements were perfomred on control cells to determine the best spermidine concentration and treatment duration. ATP and MTT assay were used as readouts (Figure 1). Based on the data obtained, a 48 hrs treatment with 0.1 ?M spermidine was subsequently used to assess different bioenergetic parameters, as well as the effects of spermidine on autophagy/mitophagy in vector versus P301L cells. Of note, since vector and P301L cells are tagged with GFP, the proliferative effect of spermidine were assessed by measuring the GFP fluorescent signal. No significant differences were observed between untreated and spermidine-treated conditions in both cell lines (p=0.638, data not shown), indicating that spermidine does not affect cell proliferation.
Cell Viability Assay
Cell viability was investigated using a MTT assay. SH-SY5Y cells were plated in at least 5 replicates into a 96-well cell culture plates at a density of 1.5 × 104 cells/well. After spermidine treatment, the cells were incubated with 5 mg/ml MTT (3-(4,5-dimethylthyazol-2-yl)-2,5-diphenyl-tetrazolium bromide) in DMEM (10 ul / well) for 2 hours. MTT is reduced to a violet formazan derivative by mitochondrial enzymatic activity.
Subsequently, the medium was removed and 200 µl of DMSO were added to each well to dissolve the formazan crystals. MTT absorbance was measured at 550 nm using the multiplate reader Cytation 3 (BioTek, Luzern, Switzerland).”
-qPCR experiments: GAPDH is know to change expression in several metabolic insults. It needs to be demonstrated that it does not change in this setting. Normalization to more than one gene is preferable. The deltaCT method was used to normalize gene expression. Is there an experimental group used as reference of is the normalization done for all across-groups measurements as reference?
Answer : We thank the referee for raising this point. We checked the raw data for changes in GAPDH expression between groups, and also following metabolic alteration using FCCP. No difference in GAPDH expression was observed between groups or with FCCP treatment (see the corresponding figure in the attached pdf document).
-The discussion misses a reflection on the limitations of the study
Answer: We agree with the referee. Discussion points related to study limitation were added. Please see:
Lines 313-317 : « Additional experiments are needed to assess the effects of spermidine on the antioxidant system (glutathione system, superoxide dismutase expression/activity), as the increase of free-radicals level is a hallmark of brain aging and neurodegenerative disorders. Studies in which oxidative damages are induced (e.g. using hydrogen peroxide) would help elucidating the effect of spermidine on oxidative stress in healthy cells. »
Lines 325-327: “Future experiments using more gentle stimuli with less mitochondrial perturbations, such as antimycin A or valinomycin may help elucidate the effect of spermidine on mitophagy in healthy cells.”
Lines 355-362: “Further studies are now needed to confirm these findings, first on other cellular models, then in vivo. Indeed, we focused here on P301L-tau because this mutation induces NFTs in mice that are similar to those observed in the brain of AD patients. The effects of spermidine should also be evaluated in the presence of other tau mutations (e.g. R406W, E10+16) involved in other tauopathies. Besides, we used here SH-SY5Y cells that are human neuron-like cells. Studies on more advanced cellular models like human induced-pluripotent stem cells (hiPSCs)-derived neurons would be more relevant to assess the effects of spermidine on the human brain physiology.”
minor:
- the abstract is not clear on the reasoning to test spermidine in a tauopathy model.
Answer : We agree with the referee. The abstract was re-written accordingly. Please see lines 11-26:
“Abnormal tau build-up is a hallmark of Alzheimer’s disease (AD) and more than 20 other serious neurodegenerative diseases. Mitochondria are paramount organelles playing a predominant role in cellular bioenergetics, namely by providing the main source of cellular energy via adenosine triphosphate generation. Abnormal tau impairs almost every aspects of mitochondrial function, from mitochondrial respiration to mitophagy. The aim of our study was to investigate the effects of spermidine, a polyamine which exerts neuroprotective effects, on mitochondrial function in a cellular model of tauopathy. Recent evidence identified autophagy as the main mechanism of action of spermidine on life-span prolongation and neuroprotection, but the effects of spermidine on abnormal tau-induced mitochondrial dysfunction have not yet been investigated. We used SH-SY5Y cells stably expressing a mutant form of human tau protein (P301Ltau mutation) or cells expressing the empty vector (control cells). We showed that spermidine improved mitochondrial respiration, mitochondrial membrane potential as well as ATP production in both control and P301Ltau-expressing cells. We also showed that spermidine decreased the level of free radicals, increased autophagy, and restored P301Ltau-induced impairments in mitophagy. Overall, our findings suggest that spermidine supplementation might represent an attractive therapeutic approach to prevent/counter-act tau-related mitochondrial impairments.”
- I suggest reporting all data as mean and SD, rather than using SEM.
Answer : We now show the data as mean and SD, as requested by the referee.
- I suggest using SI units throughout the manuscript.
Answer : We now show the SI units on the graphs.
- I see no reason to abbreviate spermidine to SPD
Answer : We thank the referee for this comment. We have removed the SPD abbreviation throughout the text.
-in table 1, measurements are relative to what?
Answer : The table 1 was removed, as the figure 2 has been modified to show to compare both cell lines (vector versus P301L cells), as well as the effect of spermidine on each measured parameter.

Round 2
Reviewer 1 Report
After careful examination of the reviewers' comments, the response of the authors, and the changes made in the manuscript, I gather that the revised version of the manuscript has addressed all the relevant concerns raised in the previous version of the paper. Hence, I endorse the final publication of the manuscript.
Reviewer 2 Report
I acknowledge that the authors replied to all my comments and took my concerns in account when revising the manuscript. The manuscript is acceptable for publication.
minor comment: the fact that experiments are done in such high glucose concentration, that is one order of magnitude larger that that in which cells are bathing in the brain, might impact the findings. I wonder if the results can be reproduced at lower, more physiological glucose concentration. That's why it's great to have substrate concentrations stated in the paper.